# Tunneling time probed by quantum shot noise

Pierre Février[1] & Julien Gabelli [1]

In typical metallic tunnel junctions, the tunneling events occur on a femtosecond timescale. An estimation of this time requires current measurements at optical frequencies and remains challenging. However, it has been known for more than 40 years that as soon as the bias voltage exceeds one volt, the junction emits infrared radiation as an electrically driven optical antenna. We demonstrate here that the photon emission results from the fluctuations of the current inside the tunneling barrier. Photon detection is then equivalent to a measurement of the current fluctuations at optical frequencies, allowing to probe the tunneling time. Based on this idea, we perform optical spectroscopy and electronic current fluctuation measurements in the far from equilibrium regime. Our experimental data are in very good agreement with theoretical predictions based on the Landauer Büttiker scattering formalism. By combining the optics and the electronics, we directly estimate the so-called traversal time.

---

[1] Laboratoire de Physique des Solides, CNRS, Université Paris-Sud, Université Paris-Saclay, 91405 Orsay, France. Correspondence and requests for materials should be addressed to J.G. (email: julien.gabelli@u-psud.fr)

Among the most intriguing questions in quantum mechanics, the traversal time $\tau_T$ that a particle takes to tunnel through a barrier has been the subject of a long-standing debate. Although several theoretical expressions have been proposed[1], the approach using a physical clock such as a Larmor's clock is now well accepted[2] and yields $\tau_T = \int_{\mathcal{F}} dz[m/(\hbar\kappa(z))]$ where $\kappa(z)$ is the magnitude of the imaginary wave vector in the classically forbidden region $\mathcal{F}$. Experimentally, this question has mainly been addressed in attosecond physics. Taking advantage of strong laser-field techniques, fundamental questions related to tunneling time in atom ionization have been clearly addressed[3–5]. As an alternative, electronic transport measurements through a tunnel junction allow for experimental control of the tunneling parameters and seem to be a natural way to investigate $\tau_T$. However, the experiments so far focused on the $I(V)$ characteristics which mainly probe the tunneling rate related to the transmission of the barrier $\mathcal{T}$. The traversal time thus only appears as a correction to the tunneling current due to the dynamical feedback of the electromagnetic environment[6,7].

An alternative way to estimate the traversal time by measuring the current–current correlations on a timescale comparable to the duration of the tunneling process. This is equivalent to measuring the current noise spectral density $S_{ii}$ at frequencies $\nu \sim 1/\tau_T$. For a typical aluminum oxide barrier, $\tau_T$ is of the order of femtoseconds and necessarily implies a measurement of $S_{ii}$ at optical frequencies. Note that for a rectangular barrier, the time for an electron to cross the barrier is given by $\tau_T = d\sqrt{m/(2(U - eV))}$ where $U$ is the barrier height, $d$ its thickness, $m$ the effective mass of electron and $V$ the bias voltage. For a common aluminum oxide barrier $U \sim 2.5$ eV, $d \sim 2$ nm and $m \sim 3.5 \times 10^{-31}$ kg which gives $\tau_T \sim 2 \times 10^{-15}$ s at 1 V. In the following, we probe the traversal time by studying the infrared radiation emitted from a macroscopic planar junction[8,9]. This emitted light is broadband with a high-frequency cut-off determined by the applied voltage $h\nu < e|V|$ and was first detected in 1976 by Lambe and McCarthy[10]. It was immediately attributed to the inelastic excitation of surface plasmon polariton (SPP) junction modes which coupled to photons due to the roughness of the metal electrodes. The tunneling current flowing through the junction is indeed accompanied by the generation of light which originates from the high-frequency component of current shot noise in the tunnel barrier[8,9]. At such voltage (~1 eV), the junction is in the far from equilibrium regime (FFER). Even if light has been observed from biased junctions prepared by electromigration[11] and from those formed between a biased scanning tunneling microscope (STM) tip and metallic sample[12–14], heating of electron gas in such nanoscopic junctions make the quantitative study of the transport properties of the junction difficult. From a theoretical point of view, the work of Lesovik and Loosen[15] could explain the reduction in light emission observed in such STM experiments as a reduction of the quantum shot noise.

In this article, we first discuss the validity of the theory given by the out-of-equilibrium generalized fluctuation dissipation relation (FDR). We show that it provides information about temporal correlations between electrons but fails to describe the dynamics of the tunneling process on the femtosecond timescale in the FFER. We then generalize the existing theoretical expression of the shot noise spectral density $S_{ii}$ to optical frequencies by using a Landauer–Büttiker (LB) scattering approach. By using electronic and optical measurements, we use the universal expression for shot noise in the radio frequency range ($\nu \ll 1/\tau_T$) to experimentally prove the validity of the tunneling limit at high bias voltage in the FFER. Finally, we demonstrate that the FDR breaks down at optical frequencies. We show that our experimental results are in very good agreement with the generalized expression of $S_{ii}$ and that they give a very good estimation of $\tau_T$, the

one-dimensional (ID; longitudinal) traversal time. This defines the unique characteristic time for the tunnel junction[16] which can thus be seen as a promising electrically driven source of SPP for future plasmonic circuits.

## Results

**Fluctuation dissipation relation.** In the tunneling limit, the noise spectral density at a finite frequency $\nu$ usually reads[17–20]:

$$S_{ii}^{(\mathrm{FDR})}(eV, h\nu) = e\{(1 + N(eV - h\nu))I(V - h\nu/e) \\ + N(eV + h\nu)I(V + h\nu/e)\}, \quad (1)$$

where $I(V)$ is the dc characteristic of the voltage-biased tunnel junction, $N(\epsilon) = 1/(\exp(\epsilon/k_B T) - 1)$ the Bose–Einstein distribution and $e$ denotes the elementary charge of electron. It is referred to as the FDR and is generally valid as shown in refs. [19,20]. The FDR is indeed in quantitative agreement with numerous experiments in the microwave regime. These experiments range from simple linear tunnel junctions[21] to tunnel junctions showing nonlinear characteristics due to superconducting electrodes[22] or to dynamical Coulomb blockade effect[23]. According to Eq. (1), the noise spectral density $S_{ii}$ vanishes for frequencies $\nu > e|V|/h$ as long as thermal fluctuations are negligible ($h\nu \gg k_B T$). This has a simple interpretation: in the wave-packet picture[24,25], the electrons are emitted from a constant voltage source with an average time separation $\tau_Q = h/e|V|$. It is thus impossible to measure current–current correlations on a timescale smaller than $\tau_Q$ (Fig. 1). $\tau_Q$ characterizes the temporal correlations between successive attempts of the electrons to cross the junction. However, if $\tau_Q$ is a characteristic time of the FDR, the traversal time $\tau_T$ is not. This means that the FDR does not provide a complete picture of the tunneling process. The reason arises from the fact that tunneling events are actually supposed to be instantaneous in the FDR (see Methods). We thus expect a breakdown of the FDR at frequency $1/\tau_T \sim \nu \lesssim 1/\tau_Q$ proving the existence of the traversal time $\tau_T$. The central result of this article

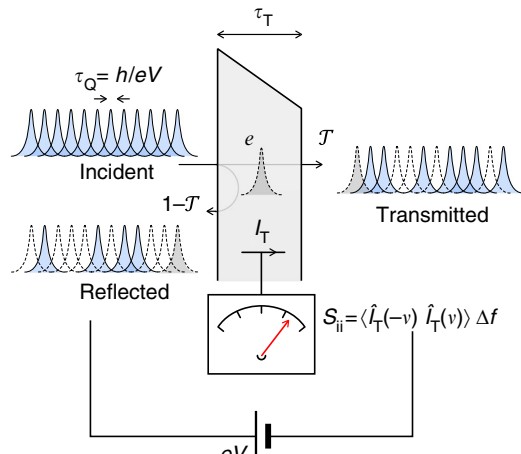

**Fig. 1** Tunnel barrier connected to a constant voltage source. According to the wave-packet picture, the voltage source acts as a regular single electron source with the average time between electrons emitted per channel per spin from the macroscopic contact is $\tau_Q = h/e|V|$. The electrons can be either transmitted through the barrier with a probability $\mathcal{T}$ or reflected back by the barrier with a probability $1 - \mathcal{T}$ giving rise to quantum partition noise. The corresponding noise spectral density $S_{ii}(\nu) \propto \left\langle \hat{I}_T(-\nu)\hat{I}_T(\nu) \right\rangle$ of the tunnel current $I_T$ then provides information about the time $\tau_T$ that an electron spends in the barrier as long as the measurement frequency $\nu$ satisfies $1/\tau_T \lesssim \nu < 1/\tau_Q$. The noise spectral density also highlights the charge accumulation which occurs in the barrier at high frequencies

is the observation of this breakdown by comparing the measured noise spectral density $S_{ii}(eV, h\nu)$ to the FDR deduced from the measured characteristics $I(V)$. This result constitutes direct evidence of the existence of a characteristic time associated with tunneling.

**Nonlinear tunneling transport.** The FFER is reached when the applied bias voltage is of the order of the tunnel barrier height $U$. In this regime, the tunneling barrier is modified by the bias voltage which leads to an intrinsic nonlinear conductance. Without a careful study of the Coulomb interactions in the tunnel barrier, gauge invariance (invariance of the current under a global voltage shift applied on both electrodes) is not systematically satisfied[26,27]. It is indeed necessary to determine the electrical potential which depends on the applied bias voltage and the possible charge accumulation in the conductor. Instead of the transfer Hamiltonian formalism, here we use the LB description to properly describe the ballistic transport through the barrier (Supplementary Note 3). Its transmission $\mathcal{T}$ is thus necessarily both energy and voltage dependent and its $I(V)$ characteristic reads:

$$I(V) = \frac{2e}{h} \int d\epsilon \mathcal{T}(\epsilon, eV)[f(\epsilon) - f(\epsilon + eV)], \quad (2)$$

where $f(\epsilon) = 1/(1 + \exp((\epsilon - \epsilon_F)/k_B T))$ is the Fermi–Dirac distribution with $\epsilon_F$ the Fermi energy. In the tunneling limit, the voltage dependence of $\mathcal{T}$ can be deduced by using the free-electron density of states and the Wentzel–Kramers–Brillouin (WKB) transmission coefficient[28] and by considering a total potential including the potential barrier $U(z)$ and the biasing energy $U_{bias}(z, V) = -eV(1 - z/d)$ as depicted in Fig. 2. Note that the effects of image charge could be considered in the potential barrier $U$: in that case, these effects would only re-normalize the barrier height. They will be taken into account only to estimate the tunneling current. It is worth emphasizing that the biasing energy is essential to explain the non-symmetric $I(V)$ characteristics depicted in Fig. 2 (Supplementary Fig. 4). We now consider the current fluctuations characterized by the non-symmetrized shot noise spectral density $S_{\alpha,\beta} = \langle \hat{I}_\alpha(-\nu)\hat{I}_\beta(\nu)\rangle \Delta f$ where $\hat{I}_\alpha(\nu)$ is the Fourier component of the current operator measured in the electrode $\alpha, \beta = L, R$ and $\Delta f$ the measurement bandwidth. For $\nu > 0$, this quantity refers to the emission quantum noise which is measured in a passive detection scheme such as the photon detector used here[15,27]. Using the scattering LB approach for a

single quantum channel of conduction in the tunneling limit ($\mathcal{T} \ll 1$), we get for $\alpha \neq \beta$[27]

$$S_{\alpha\alpha}(eV, h\nu) = \frac{2e^2}{h} \int d\epsilon \Big\{ \mathcal{T}(\epsilon - h\nu, eV)f_\alpha(\epsilon)\Big[1 - f_\beta(\epsilon - h\nu)\Big]$$
$$+ \mathcal{T}(\epsilon, eV)f_\beta(\epsilon)[1 - f_\alpha(\epsilon - h\nu)]\Big\}, \quad (3a)$$

$$S_{\alpha\beta}(eV, h\nu) = -\frac{2e^2}{h} \int d\epsilon \sqrt{\mathcal{T}(\epsilon, eV)\mathcal{T}(\epsilon - h\nu, eV)}$$
$$\Big\{ f_\alpha(\epsilon)\Big[1 - f_\beta(\epsilon - h\nu)\Big] + f_\beta(\epsilon)[1 - f_\alpha(\epsilon - h\nu)]\Big\}. \quad (3b)$$

where $f_L(\epsilon) = f(\epsilon + eV)$ and $f_R(\epsilon) = f(\epsilon)$. In the zero frequency limit and using the expression $f(\epsilon + \epsilon_0)(1 - f(\epsilon)) = N(\epsilon_0)(f(\epsilon) - f(\epsilon + \epsilon_0))$, the above expressions simplify to $S_{LL} = S_{RR} = -S_{LR} = S_{ii}^{(FDR)}$ and the FDR holds even in the nonlinear regime. However, at finite frequency, the energy and voltage dependence of the transmission $\mathcal{T}$ leads to charge accumulation in the barrier and the noise spectral density depends on the electrode where it is evaluated ($S_{LL} \neq S_{RR} \neq -S_{LR}$)[27,29]. A key question then arises: which noise spectral density is measured at optical frequencies? Following Nyquist[30], the noise spectral density $S_{ii}$ related to the radiated spectral power $P_\nu$ emitted by the tunnel junction is given by:

$$P_\nu = \mathcal{R}(\nu)S_{ii}, \quad (4)$$

where the radiation impedance $\mathcal{R}(\nu)$ stands for the coupling between the current in the conductor and the far-field-radiating electromagnetic modes. This radiation impedance takes into account the time-scales related to the plasmonic in the metallic contacts and the stemming from the optical setup. It clearly does not depend on the dc voltage and is therefore not relevant to the assessment of the traversal time. At optical frequencies, this coupling takes place in the insulating barrier because of the screening of the electromagnetic field in the metallic electrodes (Supplementary Fig. 10). An estimation of this coupling requires a microscopic description of the charge transfer. Even though it should be necessary to solve the coupled system of Schrödinger and Poisson equations to calculate the tunneling current $\hat{I}_T$, the screening in the metallic electrodes permits a simple description of $\hat{I}_T$. The bare electron inside the tunneling barrier induces a

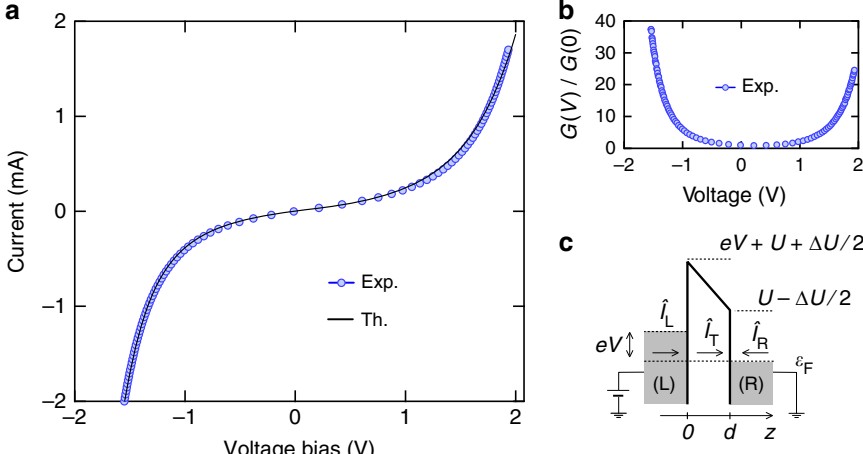

**Fig. 2** Characteristic of the tunnel junction. **a** Symbols are experimental data and solid lines are theoretical expectations of Eq. (2) using the WKB approximation on a trapezoidal barrier characterized by a mean height $U$ and an asymmetry $\Delta U$. **b** Differential conductance vs. voltage at low bias. **c** Schematic of the trapezoidal barrier modified by a bias voltage $V$

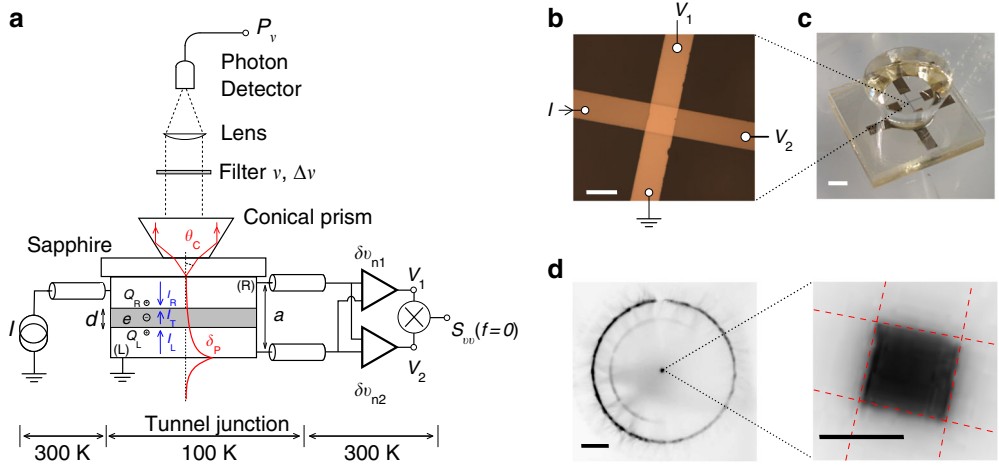

**Fig. 3** Experimental setup. **a** Schematic of the experimental setup. The blue arrows indicate the conventional direction of the current. The red curve corresponds to the amplitude of the electric field of the SPP mode. The red arrows depict the propagating rays in the Kretschmann configuration. **b** Optical micrograph of the metallic cross-junction. Scale bar: 100 μm. **c** Optical picture of the Kretschmann configuration used to couple the surface plasmon polariton localized at the interface electrode/vacuum to the radiating field. The conical prism enables to collect photons via a total internal reflection. Scale bar: 1 mm. **d** Emitted light from the tunnel junction ($I = 1.7$ mA) directly observed with a sensitive camera in the spectral range 0.4–1 μm. Scale bars: 1 mm (left) and 100 μm (zoom, right)

polarization charge $e(1 - z/d)$ and $ez/d$ in the left and right electrodes respectively (Fig. 3c). We can thus assume that the charge accumulation on the surface of the electrodes is equal on average during the tunneling event: $\hat{Q}_L = \hat{Q}_R = \hat{Q}$ with $\langle \hat{Q} \rangle = e/2$. The continuity equation $d\hat{Q}/dt = \hat{I}_L - \hat{I}_T = \hat{I}_T + \hat{I}_R$ then implies $\hat{I}_T = (\hat{I}_L - \hat{I}_R)/2$ with the conventional direction of the current (Fig. 2). The current noise spectral density $S_{ii} = \langle \hat{I}_T(-\nu)\hat{I}_T(\nu)\rangle \Delta f$ in Eq. (4) is then given by:

$$S_{ii}(eV, h\nu) = \frac{1}{4}(S_{LL} + S_{RR} - 2S_{LR}).\qquad(5)$$

Although this expression is similar to that written in refs.[14,31], it should be noted that the authors describe the charge conservation in a central dot (molecular junction) using a transfer Hamiltonian model. They are mainly interested in transfer rate and not in tunneling time. In our case, we are interested in the tunneling region described by the scattering formalism. Our formalism takes into consideration not only the effect of the Coulomb interactions in the barrier but also the time delay of the electron crossing the barrier. This is hidden in the strong energy dependence of the tunneling transmission. Under these conditions, the FDR cannot be satisfied when $\nu \sim 1/\tau_T$ (Methods).

**Experimental setup**. Our experimental setup is shown in Fig. 3c. Electronic and optical measurements are performed in a cryogenic environment at $T \sim 100$ K to prevent junction breakdown and to reduce the thermal noise in the infrared photon detector. The sample is a $100 \times 100\ \mu m^2$ planar $Al/AlO_x/Al$ tunnel junction deposited on a sapphire substrate (Fig. 3a and Supplementary Note 1). Because of the layered structure of the junction, the electromagnetic modes known as SPPs in the junction should not radiate to the free space. However, the total thickness of the junction $a \sim 10$ nm is smaller than the penetration depth of the SPP in the metal: $\delta_P = c/\omega_P \simeq 13$ nm with $\omega_P = 14.7$ eV the plasma frequency of aluminum. This enables the coupling between the SPP mode localized at the interface electrode/vacuum (Fig. 3c) and the propagating mode in the substrate (Supplementary Figs. 8–11). This corresponds to the

Kretschmann configuration where the coupling appears at a specific angle $\theta_P \simeq \arcsin(1/n) \simeq 35°$ where $n$ stands for the refractive index of sapphire[32]. We use total internal reflection in a conical prism to collect the emitted photons (Fig. 3b, c). The current noise $S_{ii}(eV, h\nu)$ at an optical frequency $\nu$ is measured at two different frequencies corresponding to the wavelengths $\lambda = c/\nu = 0.9 \pm 0.02\ \mu m$ and $1.3 \pm 0.015\ \mu m$. The current noise at zero frequency $S_{ii}(eV, h\nu = 0)$ is measured at radio frequencies with a standard cross-correlation technique (Supplementary Note 2 and Supplementary Figs. 1–3).

**Noise measurement in the radio frequency range of 20 to 100 kHz**. At high voltage, the $I(V)$ characteristic shown in Fig. 2 exhibits a strong nonlinearity: the differential resistance varies by more than one order of magnitude ranging from 6 kΩ at low bias to 150 Ω at high bias. From the theoretical expectation of Eq. (2) and using the WKB approximation assuming an homogenous trapezoidal barrier[33] to evaluate the transmission of the tunnel junction, we estimate the mean barrier height $U \sim 2.7$ eV, its asymmetry $\Delta U \sim 2.9$ eV and its thickness $d \sim 2$ nm (lower inset of Fig. 2 and Supplementary Figs. 5 and 7). Although the junction is symmetrical (identical electrodes), the strong asymmetry of the barrier can be explained by the roughness of the thin electrodes as has already been reported in the literature[34]. The thickness $d$ is in agreement with the capacitance of the junction ~0.5 nF.

In the FFER, the bias voltage is of the order of the tunnel barrier height and it is legitimate to wonder if the tunneling limit is still valid and if the traversal time still has a meaning. As previously discussed, the FDR is universal at zero frequency and may be used to test the tunneling regime[35]. Current fluctuations $S_{ii}(eV, h\nu = 0)$ are measured in the radio frequency range with low noise voltage amplifiers giving access to voltage fluctuations $S_{vv} = g(|Z_{setup}(eV)|^2 S_{ii} + S_{vv,setup}(eV))$ where $g$ is the global gain of the amplifier chain, $Z_{setup}$ is the transimpedance of the measurement setup and $S_{vv,setup}$ its excess noise. Because of the large variation of the tunneling resistance, a careful calibration is required to extract the current noise $S_{ii}$. The voltage-dependent transimpedance $Z_{setup}$ and the excess noise $S_{vv,setup}$ are determined by using an external noise source while $g$ is deduced from the measurement of the shot noise in the linear regime (Supplementary Note 2). Figure 4 shows the current noise $S_{ii}$ in

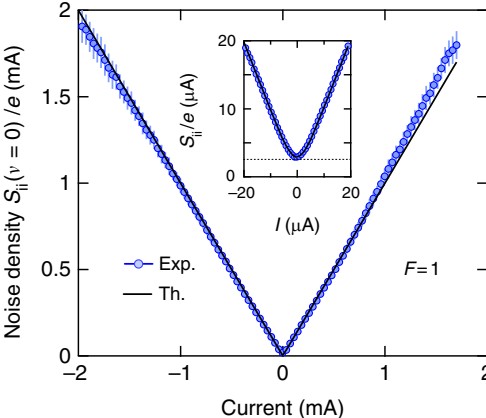

**Fig. 4** Electronic shot noise of the tunnel junction. Measurements done in the bandwidth 20–100 kHz. The Fano factor $F = 1$ characterizes the tunneling regime. The error bars are SD. Inset: Zoom at low bias voltage. Solid lines correspond to the theoretical expectation of Eq. (6) with $T = 100$ K

the FFER. Although the tunnel resistance is strongly nonlinear, $S_{ii}$ clearly satisfies the FDR at zero frequency,

$$S_{ii}(eV, h\nu = 0) = \frac{eI(V)}{\tanh(eV/2k_BT)}. \quad (6)$$

In the high bias limit $eV \gg k_BT$, the current noise is linearly proportional to the dc current. The Fano factor $F = S_{ii}(eV, h\nu = 0)/(e|I|)$ is then equal to one and confirms that electronic transport through the junction operates in the tunneling limit even at high bias voltage. This rules out the presence of pinholes in the barrier since it would imply a significant reduction of the shot noise with $F < 1$. Let us assume that the static resistance decreases from $R = 6$ kΩ to $R = 0.85$ kΩ because of the appearance of $N$ pinholes with an average transmission $\mathcal{T}_p \sim 1$. This would imply $N\mathcal{T}_p \simeq 13$ at high bias voltage. The Fano factor related to the parallel association of tunneling channels and pinholes would then be $F = 1 - \frac{2R}{R_K}N\mathcal{T}_p^2$, with $R_K = h/e^2 \simeq 25.8$ kΩ the quantum of resistance. Finally, $F \simeq 1 - 0.85\mathcal{T}_p \sim 0.15$, this is not what we observe. We notice systematic errors at high positive bias. They cannot be attributed to Joule heating since they should also be observed for negative bias. The fact that the calibration is off by ~10% is attributed to parasitic capacitances of the measurement setup which are not included in $Z_{setup}$.

**Noise and traversal time estimation at optical frequencies.** Figure 3d shows an image of the light emission pattern from the tunnel junction when the camera is focused on the conical prism. In the center, a small amount of light comes directly from the tunnel junction (zoom in Fig. 3d). This is due to the surface roughness of the electrodes allowing SPP scattering at the surface of the upper electrode[36]. The homogenous light intensity indicates that the electron-to-photon conversion in the tunnel junction is also homogenous over the surface of the junction confirming the absence of pinholes in the barrier. However, the bright ring in Fig. 3d reveals that more than 98% of the light is emitted at the specific angle $\theta_p$ as expected in the Kretschmann configuration (Supplementary Note 6). The light power $P_\nu\Delta\nu$ is plotted as a function of the bias voltage for two different wavelengths in Fig. 5a. The inset of Fig. 5a displays the relationship between the light power for the two wavelengths on a log-log plot[37]. Data points do not fit the black-body radiation law (solid

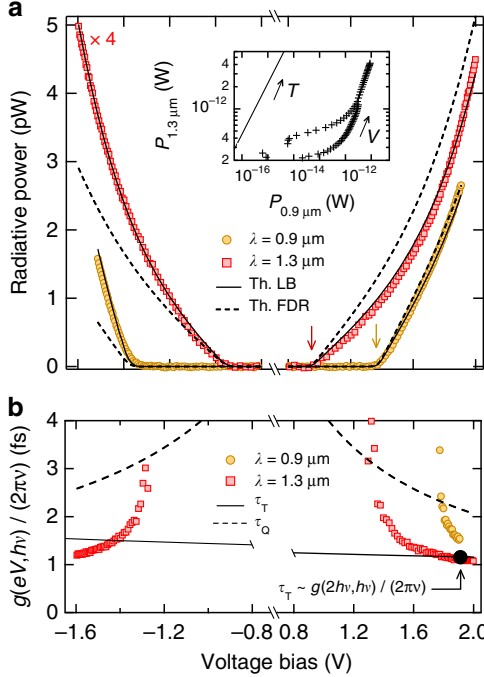

**Fig. 5** Measurements at optical frequencies. **a** Radiated power as a function of bias voltage $V$ in the configuration depicted in Fig. 3. Markers correspond to data recorded at different wavelengths. Data at $\lambda = 1.3$ µm (red squares) have been multiplied by a factor 4 for clarity. Vertical arrows define the voltage cut-off $eV = hc/\lambda$. Solid lines and dashed lines are theoretical expectations from Eqs. (1) and (5) respectively. Horizontal axis has been split for clarity. Inset: Relationship between the light power at two different wavelengths with increasing bias voltage. Solid line corresponds to the black-body law with increasing temperature. **b** Experimental data of $g(eV, h\nu)/(2\pi\nu)$ recorded at different wavelengths (markers). The dashed lines correspond to the correlation time $\tau_Q$ whereas the solid line reads for the 1D traversal time at Fermi energy $\tau_T(eV, h\nu)$. Assuming that $\tau_T$ is rather constant in the energy range $[\epsilon_F, \epsilon_F + h\nu]$ we are interested in the measured value of the 1D traversal time at $\epsilon \sim h\nu$ given by the black dot $\tau_T \sim \frac{g(2h\nu, h\nu)}{2\nu}$ (Methods)

line in inset) and, as previously mentioned, the Joule heating cannot be responsible for the observed photon emission. As already observed by Lamb and McCarthy[10], the light power exhibits a sharp voltage cross-over at $e|V| = hc/\lambda$: electrons crossing the tunnel junction relax their energy by emitting photons at a frequency $\nu \leq e|V|/h$. This cross-over is predicted by both the FDR and the LB theories. However, our data clearly disagree with the FDR (dashed line in Fig. 5a) and are in very good agreement with the LB relation of Eq. (5) (solid line in Fig. 5a). This proves the existence of the traversal time. The LB approach explains the bias-polarity dependence of the light emission which has previously been observed but not justified[36,38,39]. It also allows one to extract the radiation impedance according to Eq. (4). This gives $\mathcal{R}(\lambda = 0.9$ µm$) = 2.5 \pm 0.1$ mΩ and $\mathcal{R}(\lambda = 1.3$ µm$) = 2.3 \pm 0.1$ mΩ, about a factor of four higher than our rough estimation in the limit $\nu/\nu_p \ll 1$[9]. Unlike the low frequency noise which is bonded by the $RC$ frequency cut-off (1/($2RC$) ~ 100 kHz), the spectral power density at optical frequencies involves the radiation impedance $\mathcal{R}(\nu)$ which does not exhibit any high-frequency cut-off (Supplementary Note 6)[40]:

$$\mathcal{R}(\nu) = \frac{1}{\beta n^5}\left(\frac{d}{\delta_p}\right)^2\left(\frac{\nu}{\nu_p}\right)^3 Z_{vac} \underset{\lambda=1\mu m}{\sim} 0.5 \text{ mΩ}, \quad (7)$$

where $\beta = \tanh(a/\delta_P) \simeq 0.69$, $v_P = \omega_P/2$, $n = 1.75$ is the refractive index of sapphire and the alumina dielectric barrier, $d \simeq 2$ nm is the thickness of the barrier and $Z_{vac} \simeq 376\,\Omega$ is the vacuum impedance. This under-estimation can not only be attributed to the approximative values of the thickness and the refractive index of the dielectric barrier but also to the interband transition at $\lambda_{inter} = 0.825\,\mu$m in aluminum. The $0.1\,m\Omega$ uncertainty on radiative resistance $\mathcal{R}(\nu)$ comes from the fit of the data but one has to keep in mind that an error on the detection efficiency could induce a systematic error of about 20%. We assume here that the coupling between the current fluctuations and the electric field takes place in the insulating barrier. This is justified by the screening of the electric field in the metal. If we only consider the coupling in the electrodes, we should expect a radiation impedance in the $\mu\Omega$ range, three orders of magnitude smaller than the observed one (Supplementary Note 6).

The shot noise spectral density at optical frequencies gives direct information about the current fluctuations on the femtosecond timescale. By using the scattering LB approach for the two-dimensional tunnel barrier, we obtain an estimation of the one-dimensional traversal time given by the WKB approximation (Methods):

$$\tau_T \sim \frac{g(2h\nu, h\nu)}{2\pi\nu} \quad \text{with}$$
$$g(eV, h\nu) = \log\left(1/(4\tilde{F}-1) + \sqrt{1/(4\tilde{F}-1)^2 - 1}\right), \tag{8}$$

where $\tilde{F} = S_{ii}(eV, h\nu)/(e|I(V)|)$ is a voltage and frequency-dependent Fano-like factor. It then only involves measurable quantities $S_{ii}(eV, h\nu)$ and $I(V)$ where $S_{ii}$ is deduced from the light power and the value of the radiation impedance previously determined. Figure 5b shows $g(eV, h\nu)/(2\pi\nu)$ as a function of the bias voltage for the two different wavelengths. This expression is only defined for sufficiently high voltage when $1/4 < \tilde{F} \leq 1/2$. However, the bias voltage has to be kept smaller than 2 V to avoid damaging the junction. The estimation $\tau_T \sim g(2h\nu, h\nu)/(2\pi\nu) \sim 1.1$ fs can thus only be deduced from the measurement at $\lambda = 1.3\,\mu$m with $eV = 2h\nu = 1.9$ V. This value is in very good agreement with the theoretical one-dimensional traversal times calculated for a trapezoidal barrier in the energy range $[\epsilon_F, \epsilon_F + h\nu]$ (Methods).

## Discussion

We have experimentally demonstrated that the quantum shot noise spectral density cannot be simply expressed by the universal FDR. If this discrepancy is formally due to both energy and voltage dependence of the transmission $\mathcal{T}$ (Methods), the key reason is that the charge inside the tunneling barrier fluctuates and this reflects the fact that electrons spend a certain time inside the barrier. To take into account this time spent in the barrier, we have developed a theory using the formalism of LB which also gives insights on the mechanism at the origin of the optical radiation emitted by biased planar tunnel junction. The radiation impedance in the $m\Omega$ range is indeed small and appears as a central quantity in the understanding of the poor light emission efficiency of tunnel junctions. This leads us to redefine the efficiency with respect to the dissipated Joule power: $\eta = \int_0^{+\infty} P_\nu d\nu/(V \times I) \sim 4 \times 10^{-8}$. According to this definition, we can show that the efficiency is now directly related to the radiation impedance: $\eta \sim \eta_0 \mathcal{R}(eV/h)/R_K$ where $R_K = h/e^2 \simeq 25.8$ k$\Omega$ is the quantum of resistance and $\eta_0 \simeq 0.047$ is a constant slightly dependent on the details of the barrier (Supplementary Fig. 12). We emphasize that this definition contrasts with the usual one which is given by the electron-to-photon conversion rate. We find the former more appropriate since it reflects the fact that, in metallic tunnel junctions, electrons with energy smaller

than bias voltage can contribute to the current. Unlike semiconductors, the lack of a band gap in metals indeed implies that each electron crossing the barrier emits a bunch of photons in a spectral range $0 < \nu < e|V|/h$ with a radiated spectral power $P_\nu$ proportional to the current. The emitted light power is then proportional to the Joule power $V \times I$.

Regarding the mechanism of photon emission in metallic tunnel junctions, it is usually attributed to the spontaneous emission in the barrier due to inelastic electron tunneling. Although it has recently been demonstrated in a Van der Waals quantum tunnel junction[41] that part of the emission processes is due to direct photon emission, usually a two-step process is assumed with tunneling: electrons inelastically excite SPP which then couple to photons. The inelastic tunneling process is characterized by an electron-to-plasmon conversion rate which roughly describes the coupling to the electromagnetic environment[16,42–44]. In our case, we give a microscopic description of the tunneling by using the LB scattering approach. It is worth noting that the LB approach which is used here only considers elastic tunneling processes. In this description, the energy relaxation formally takes place in the electrodes and corresponds to electron–hole pair recombination specified by $S_{LL}$, $S_{RR}$ and $S_{LR}$[29]. Nevertheless, by considering the coupling to the electric field only in the dielectric layer, we implicitly assume a relaxation in the tunneling barrier associated with the noise spectral density $S_{ii} = (S_{LL} + S_{RR} - 2S_{LR})/4$. Thus, our approach is not in contradiction with the inelastic interpretation. More precisely, by only considering the elastic tunneling current to calculate the radiation impedance, we just neglect the feedback of the electromagnetic environment on the tunneling processes. This feedback, known as dynamical Coulomb blockade, is responsible for a correction of the current of the order of the ratio $\mathcal{R}(\nu)/R_K$[45–47]. In our case, this effect is negligible, which validates a posteriori our approach using the elastic current.

In conclusion, we have measured the current fluctuations $S_{ii}$ in a metallic tunnel junction in the optical domain and deduced an estimation of the traversal time $\tau_T \sim 1.1$ fs. In this regime, $S_{ii}$ may no longer be described by the usual fluctuation dissipation relation because of the energy and voltage dependence of the tunneling transmission. We have shown how this dependence can be incorporated into the LB formalism to ensure the gauge invariance of the $I(V)$ characteristics in the far-from-equilibrium regime and thus describe the quantum fluctuations of the current at optical frequencies. This theoretical description is in good agreement with our experimental results, it allows us to give an approximative value of the traversal time by a fitting-free model and sheds light on the estimation of the quantum efficiency of metallic tunnel junction as a light emitter. Our experimental approach demonstrates that optical measurements are a powerful tool to study the quantum electronic transport at high energy (~1 eV). Such measurements extend the range of applicability of mesoscopic electronic transport. The traversal time, which is the average time that the particle spends in the barrier[48], could for instance be compared to the average time given by the electric waiting time distribution which has been recently calculated in mesoscopic conductors[49,50].

## Methods

**Planar tunnel junction.** In the article, for the sake of simplicity, the LB formalism (Eqs. (2), (3a) and (3b)) is only considered for a 1D single channel of conduction. We demonstrate in the supplemental material that the three-dimensional (3D) case can be deduced by summing over all the transversal modes, and the tunnel junction is then equivalent to the parallel association of the $M = S/\lambda_F^2$ transversal modes of conduction contained in the tunnel junction area $S$ with a mean transmission:

$$\mathcal{T}(\epsilon, eV) = \int_0^\epsilon T_{WKB}(\epsilon - \epsilon_\perp, eV)\frac{d\epsilon_\perp}{\epsilon_F}, \tag{9}$$

where $T_{\mathrm{WKB}}\left(\epsilon_{\|}=\epsilon-\epsilon_{\perp},eV\right)$ is the WKB transmission through a 1D homogeneous barrier and $\lambda_{\mathrm{F}}$ the Fermi wavelength. However, we will see in the following that the 3D case can also be obtained by an integration over the longitudinal energy $\epsilon_{\|}=\epsilon-\epsilon_{\perp}$. Using this integration, we will shown that the Fano-like factor at optical frequencies directly gives the 1D traversal time $\tau_{\mathrm{T}}$ without any assumption about the dimensionality of the junction.

**Shot noise spectral density and the FDR**. For a 1D tunneling barrier, the tunneling transmission is related to the traversal time $\tau_{\mathrm{T}}$ according to (Supplementary Note 4):

$$\frac{T_{\mathrm{WKB}}(\epsilon_2,eV)}{T_{\mathrm{WKB}}(\epsilon_1,eV)}=\exp\left\{\frac{2}{\hbar}\int_{\epsilon_1}^{\epsilon_2}\tau_{\mathrm{T}}(\epsilon',eV)d\epsilon'\right\}. \tag{10}$$

Using the scattering LB approach in the tunneling limit ($\mathcal{T}\ll 1$), the integration over the longitudinal energy $\epsilon_{\|}$ of the 2D versions of Eqs. (3a) and (3b) gives:

$$S_{\mathrm{ii}}(eV,h\nu)=\frac{2e^2}{h}\left\{(1+N(eV-h\nu))\int d\epsilon_{\|}\left(\frac{\sqrt{T_{\mathrm{WKB}}(\epsilon_{\|},eV)}+\sqrt{T_{\mathrm{WKB}}(\epsilon_{\|}+h\nu,eV)}}{2}\right)^2\right.$$
$$\left[\tilde{f}\left(\epsilon_{\|}\right)-\tilde{f}\left(\epsilon_{\|}+e\left(V-\frac{h\nu}{e}\right)\right)\right]+N(eV+h\nu)$$
$$\left.\int d\epsilon_{\|}\left(\frac{\sqrt{T_{\mathrm{WKB}}(\epsilon_{\|},eV)}+\sqrt{T_{\mathrm{WKB}}(\epsilon_{\|}-h\nu,eV)}}{2}\right)^2\left[\tilde{f}\left(\epsilon_{\|}\right)-\tilde{f}\left(\epsilon_{\|}+e\left(V+\frac{h\nu}{e}\right)\right)\right]\right\} \tag{11}$$

where $\tilde{f}\left(\epsilon_{\|}\right)=-\frac{Mk_{\mathrm{B}}T}{\epsilon_{\mathrm{F}}}\ln\left(f\left(-\epsilon_{\|}\right)\right)$ is a quasi-distribution and corresponds to the integration of $f\left(\epsilon_{\perp}+\epsilon_{\|}\right)$ over $\epsilon_{\perp}$. Note that no approximations have been used so far and we have chosen here to integrate over $\epsilon_{\|}$ instead of $\epsilon_{\perp}$. $T_{\mathrm{WKB}}\left(\epsilon_{\|},eV\right)$ corresponds here to the one dimensional WKB transmission. Then, the FDR reads:

$$S_{\mathrm{ii}}^{(\mathrm{FDR})}(eV,h\nu)=\frac{2e^2}{h}\left\{(1+N(eV-h\nu))\int d\epsilon_{\|}\,T_{\mathrm{WKB}}\left(\epsilon_{\|},eV-h\nu\right)\right.$$
$$\left[\tilde{f}\left(\epsilon_{\|}\right)-\tilde{f}\left(\epsilon_{\|}+e\left(V-\frac{h\nu}{e}\right)\right)\right]+N(eV+h\nu)$$
$$\left.\int d\epsilon_{\|}\,T_{\mathrm{WKB}}\left(\epsilon_{\|},eV+h\nu\right)\left[\tilde{f}\left(\epsilon_{\|}\right)-\tilde{f}\left(\epsilon_{\|}+e\left(V+\frac{h\nu}{e}\right)\right)\right]\right\}. \tag{12}$$

Equations (11) and (12) are equivalent only if $T_{\mathrm{WKB}}\left(\epsilon_{\|},eV\right)+\sqrt{T_{\mathrm{WKB}}\left(\epsilon_{\|}\pm h\nu,eV\right)}=2\sqrt{T_{\mathrm{WKB}}\left(\epsilon_{\|},eV\mp h\nu\right)}$. The FDR thus holds at finite frequency only if the transmission is of the form (Supplementary Note 5):

$$T_{\mathrm{WKB}}\left(\epsilon_{\|},eV\right)=T_0\left(1+\frac{\epsilon_{\|}-eV/2}{\epsilon_0}\right)^2. \tag{13}$$

This form is only a good approximation of the tunneling transmission at small bias voltage $eV\ll U$ with:

$$T_0=\exp\left(-\sqrt{\frac{8mUd^2}{\hbar^2}}\right)\quad\text{and}\quad\epsilon_0=\sqrt{\frac{2\hbar^2 U}{md^2}}. \tag{14}$$

In this limit, $\tau_{\mathrm{T}}=\sqrt{md^2/(2U)}$ and $h\nu\lesssim eV\ll U$ implies $\nu\tau_{\mathrm{T}}\ll-(\ln\mathcal{T}_0)/8<1$ which corresponds to a negligible traversal time.

**Traversal time estimation**. According to Eqs. (10) and (11) and assuming that $\tau_{\mathrm{T}}$ is rather constant on the energy scale $[\epsilon_{\mathrm{F}},\epsilon_{\mathrm{F}}+h\nu]$, the zero temperature limit $k_{\mathrm{B}}T\ll eV,h\nu$ gives (Supplementary Note 4 + Supplementary Fig. 6):

$$S_{\mathrm{ii}}(eV,h\nu)\simeq\frac{2e^2}{h}\left\{\Theta(eV-h\nu)\left(\frac{1+e^{2\nu\tau_{\mathrm{T}}}}{2}\right)^2\int d\epsilon_{\|}\,T_{\mathrm{WKB}}\left(\epsilon_{\|},eV\right)\right.$$
$$\left[\tilde{f}\left(\epsilon_{\|}\right)-\tilde{f}\left(\epsilon_{\|}+e\left(V-\frac{h\nu}{e}\right)\right)\right]-\Theta(eV+h\nu)\left(\frac{1+e^{-2\nu\tau_{\mathrm{T}}}}{2}\right)^2 \tag{15}$$
$$\left.\int d\epsilon_{\|}\,T_{\mathrm{WKB}}\left(\epsilon_{\|},eV\right)\left[\tilde{f}\left(\epsilon_{\|}\right)-\tilde{f}\left(\epsilon_{\|}+e\left(V+\frac{h\nu}{e}\right)\right)\right]\right\},$$

with $\Theta$ the Heaviside step function. Similarly, the LB formula Eqs. (2) and (10) give at the special bias voltages $V=\pm h\nu/e$:

$$I(\pm h\nu/e)=\frac{2e}{h}\int d\epsilon_{\|}\left(T_{\mathrm{WKB}}\left(\epsilon_{\|},\pm 2h\nu\right)+T_{\mathrm{WKB}}\left(\epsilon_{\|}\right.\right.$$
$$\left.\mp h\nu,\pm 2h\nu\right)\left[\tilde{f}\left(\epsilon_{\|}\right)-\tilde{f}\left(\epsilon_{\|}\pm h\nu\right)\right]$$
$$\simeq\frac{2e}{h}\left(1+e^{\pm 4\nu\tau_{\mathrm{T}}}\right)\int d\epsilon_{\|}\,T_{\mathrm{WKB}}\left(\epsilon_{\|},\right. \tag{16}$$
$$\left.\pm 2h\nu\right)\left[\tilde{f}\left(\epsilon_{\|}\right)-\tilde{f}\left(\epsilon_{\|}\pm h\nu\right)\right].$$

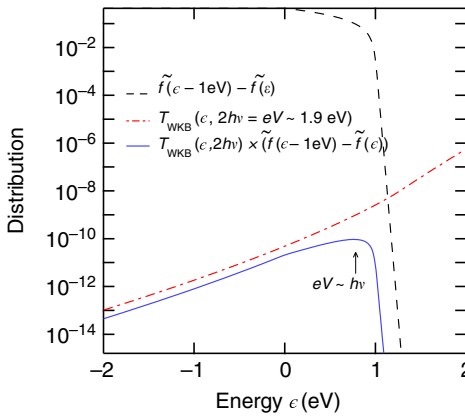

**Fig. 6** Traversal time estimation. The dashed black curve is the difference between the distribution functions appearing in Eqs. (15) and (16) for 1 eV. The dash-dotted red curve represents the transmission $T_{\mathrm{WKB}}$. The product of these two functions is pictured in blue (solid line): this broad-peaked curve allows us to evaluate the generalized Fano factor, and thus the traversal time in the zero temperature limit

Figure 6 shows the product of the distribution function $\left[\tilde{f}\left(\epsilon_{\|}\right)-\tilde{f}\left(\epsilon_{\|}\pm h\nu\right)\right]$ and the transmission $T_{\mathrm{WKB}}(\epsilon,\pm 2h\nu)$ appearing in Eqs. (15) and (16). This distribution function is a broad peak spanning from $\epsilon_{\mathrm{F}}$ to $\epsilon_{\mathrm{F}}+h\nu$. Thus, the generalized Fano factor $\tilde{F}(eV,h\nu)=S_{\mathrm{ii}}(eV,h\nu)/(e|I(V)|)$ evaluated at $V=\pm 2h\nu/e$ can be expressed in the zero temperature limit as a function of $\tau_{\mathrm{T}}(\epsilon\sim h\nu/2,eV=2h\nu)$:

$$\tilde{F}(\pm 2h\nu,h\nu)=\frac{S_{\mathrm{ii}}(\pm 2h\nu,h\nu)}{e|I(\pm 2h\nu/e)|}\simeq\frac{(1+e^{\pm 2\nu\tau_{\mathrm{T}}})^2}{4(1+e^{\pm 4\nu\tau_{\mathrm{T}}})}. \tag{17}$$

Finally, by inverting Eq. (17), we get:

$$\tau_{\mathrm{T}}\simeq\frac{1}{2\nu}\ln\left(1/(4\tilde{F}-1)+\sqrt{1/(4\tilde{F}-1)^2-1}\right). \tag{18}$$

Note that the estimated traversal time corresponds to the 1D motion through the barrier.

## Data availability
The data that support the findings of this study are available from the corresponding author on reasonable request.

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

## Acknowledgements

We acknowledge fruitful discussions with M. Aprili, J. Basset, E. Boer-Duchemin, J. Estève, J-J Greffet, E. Pinsolle, F. Portier, B. Reulet, I. Safi and P. Simon. We also thank A. Crépieux for useful insight. This work was funded by ANR-11-JS04-006-01, Investissements d'Avenir LabEx PALM (ANR-10-LABX-0039-PALM) and ANR-15-CE24-0020.

## Author contributions

J.G. designed the project and conceived the experiment. Both authors, P.F and J.G, fabricated the device, performed the measurements, analyzed the data and interpreted the results. J.G. wrote the manuscript.

## Additional information

**Competing interests:** The authors declare no competing interests.

