## [Peer Review File · Nature Communications]

Reviewers' comments:

Reviewer #1 (Remarks to the Author):

[none]

Reviewer #2 (Remarks to the Author):

This manuscript reports on an experimental demonstration of the finite tunneling time of electrons in a metallic tunneling junction device. To my knowledge, these traversal times have not been detected in an electronic device before. In order to read out these extremely short timescales, a combination of optical and electronic methods is exploited.

I find that this work is very rigorous, combining theoretical and experimental treatments.

With this experiment, the authors reach a window of time-scales to which measurements in electronics have up to now mostly been blind.

This research thus gives new insights into this range of electronic timescales, which is relevant both for fundamental as well as for applicational reasons.

I have the following questions and remarks:

- I find that the way in which the authors introduce the tunneling time is well-grounded. However, it would be good, if the authors could better explain how they can exclude that other time-scales could play a role here, for example stemming from the optical setup or fluctuations within the metallic contacts.

- I would ask the authors to include the following reference, which is directly related to their work - even though in a very different system:

P. Eckle, A. N. Pfeiffer, C. Cirelli, A. Staudte, R. Dörner, H. G. Müller, M. Büttiker, U. Keller: Attosecond Ionization and Tunneling Delay Time Measurements in Helium. *Science* 322, 1525 (2008).

- In Fig. 5, all plots are asymmetric with respect to voltage reversal. Can you comment on this? In particular the measurement curve for $\lambda=0.9\mu\text{m}$ is only measured for one voltage direction and the theoretical estimate for the traversal times is very different.

- In the conclusion, the authors claim that they "directly measure the traversal time". However, I think that this measurement is very much based on an underlying complex model and therefore relies on assumptions that were made. It is not clear to me whether this "direct" really represents what has been done here.

A few smaller remarks about possible improvements of the manuscript:

- There is a number of places where language could be improved. See for example "peculiar" after Eq. (14) which should rather be "specific"? There are also some small grammar mistakes in the text.

- In Fig. 1 the authors call their setup a turnstile. This is a bit misleading, since a turnstile leads to a clocked transmission due to a time-dependent driving.

Reviewer #3 (Remarks to the Author):

In the present manuscript by Fevrier and Gabelli show that photon emission resulting from current fluctuations inside a tunneling barrier can be used to probe the tunneling time. This bases on the idea that the photon emission is equivalent to the current fluctuations measurement at optical frequencies. Thereby, they perform optical spectroscopy measurement as well as electronic transport measurements at biased tunneling junctions, i.e., in the regime far from equilibrium. The experimental data are in nice agreement with theoretical predictions based on the Landauer-Büttiker scattering formalism underlining the relevance of their approach. This approach enables a direct measurement of the traversal-time, i.e., the traversal-time can be extracted.

Generally speaking, this allows the authors to estimate the quantum efficiency of metallic tunnel junctions as a light emitter. This approach demonstrates that optical measurement can be used to study the quantum electronic transport at elevated energy (~ 1 eV). The analysis of the traversal-time, i.e., the average time that the electrons spends in the barrier are of general interest and may be used to shed further light, e.g., on the transport properties of mesoscopic conductors.

I find the manuscript interesting to read while the authors are using a quite specialized technique to gain insight into transport phenomena in tunneling junctions. The paper is competently written and the given explanations are scientifically sound. However, for the readership of Nature Communications the given explanations appear quite complex and in the present form it may be better suited for a more specialized audience. Thus, I may only recommend publishing the manuscript in Nature Communications if the authors can enhance the accessibility of the manuscript also for non-specialists in the field.

Reviewer #4 (Remarks to the Author):

The manuscript reports measurements of transport and photon emission from a planar tunnel-junction of few nm width at high voltage compared to the barrier height of the order volts.

It is shown how the data for photon emission and current vs. voltage can be fitted to the current noise

in a single-channel transport model (trapezoidal tunnel barrier model). It is pointed out how the Fluctuation-Dissipation relation does not fit the results well. This does not come as a surprise since this is a highly non-equilibrium situation.

From the data the traversal-time is estimated.

This latter point seems to be new to me and may be an important step forward. However, I have some doubts w.r.t. the theoretical model

and to what extent the traversal-time estimate hinge critically on this model. Moreover, is this work providing a substantial contribution beyond the more STM-type experiments (e.g. PRL 109, 186601 (2012), not cited) where the channels are defined by a few molecular orbitals? I am not sure.

1) The barrier is fabricated as symmetric L-R. Although this is the case, the model have to include a very large asymmetry parameter ΔU on the order of the barrier itself, U to fit the experiments. On the other hand the charges on the L, R electrodes are assumed to be the same. This would seem reasonable if there were no signatures of asymmetry but this is not the case.

2) The model assumes completely planar surfaces such that the roughness is on a larger scale than the Fermi wavelength and transverse channels can be defined by splitting the free-electron energy expression. Is this justified? It is stated that there are no pin-holes but there could be roughness which might explain the high asymmetry cf. point 1.

3) The values of U , and ΔU should be justified. Are these reasonable/understandable based on the materials?

4) It is unclear to me that the traversal time generalizes from 1D to 3D in this way. Do you have a reference for the multi-channel case?

5) The value of the traversal time estimate (1.1 fs) is on the very low part of the expected values for the model (Fig. 5). Is it clear why this is not inside the expected area (gray)?

Dear Reviewers,

We thank reviewers for their enthusiastic opinions and constructive comments. We have addressed all points raised by the reviewers, and we believe this has significantly improved the manuscript. The corrections made to the text are **marked in red** and the removed sentences are ~~crossed out~~, for clarity. Our detailed point-by-point answer follows.

1 Answers to Reviewer #1

No comment has been done, no question has been asked.

2 Answers to Reviewer #2

We are glad he/she acknowledges that our work addresses “ a window of time-scales to which measurements in electronics have up to now mostly been blind ”. He/she provided detailed comments and suggestions to improve the manuscript. Below are our answers to his/her suggestions.

1. I find that the way in which the authors introduce the tunneling time is well-grounded. However, it would be good, if the authors could better explain how they can exclude that other time-scales could play a role here, for example stemming from the optical setup or fluctuations within the metallic contacts.

The traversal-time is deduced from noise spectral density S_{ii} at optical frequencies according to Eq. (4):

$$P_\nu = \mathcal{R}(\nu)S_{ii}, \quad (1)$$

where the radiation impedance $\mathcal{R}(\nu)$ stands for the coupling between the current in the conductor and the far field radiating electromagnetic modes. This impedance takes into account the time-scales related to the plasmonic in the metallic contacts and the stemming from the optical setup. It is dc voltage independent whereas our estimate of the traversal-time is based on voltage dependence of the electronic quantity S_{ii} . Note that the measured S_{ii} shows, as expected, a sharp threshold at $|V| = h\nu/e$ which confirms that the measured quantity is related to electronic transport. Manuscript has been changed in lines 94-96.

2. I would ask the authors to include the following reference, which is directly related to their work - even though in a very different system: P. Eckle, A. N. Pfeiffer, C. Cirelli, A. Staudte, R. Dörner, H. G. Muller, M. Büttiker, U. Keller: Attosecond Ionization and Tunneling Delay Time Measurements in Helium. *Science* 322, 1525 (2008).

We agree with the reviewer that the tunneling-time can be introduced by citing the most convincing experiment measuring it. These experiments concern the tunneling time in atom ionization. We have changed the introduction (lines 21-24) and added references [3,4,5] in the new manuscript.

3. In Fig. 5, all plots are asymmetric with respect to voltage reversal. Can you comment on this? In particular the measurement curve for $\lambda = 0.9\mu m$ is only measured for one voltage direction and the theoretical estimate for the traversal times is very different.

The plot are asymmetric because of the strong asymmetry of the tunnel barrier (see lines 134-138 in the new manuscript, lines 78-79 in the new supplementary materials and response 2 to the Reviewer #4). It follows that we can only assess the value of the traversal time for $\lambda = 1.3\mu m$ at the positive voltage $V = 2h\nu \simeq 1.8 V$ otherwise the junction may be damaged (see lines 189-190 in the new manuscript).

4. In the conclusion, the authors claim that they “directly measure the traversal time”. However, I think that this measurement is very much based on an underlying complex model and therefore relies on assumptions that were made. It is not clear to me whether this “direct” really represents what has been done here.

We agree with the reviewer that the term “direct measure” was inappropriate. This has been corrected in line 55 as well as in the calculations of the Methods. Moreover, we have added a remarque in lines 184-186 to emphasize that this estimate only depends on measurable quantities. In the previous version of the manuscript, we have used the single mode transmission $\mathcal{T} = M/\epsilon_F \int_0^\epsilon T_{WKB}(\epsilon - \epsilon_{\parallel}, eV) d\epsilon_{\parallel}$ to estimate the traversal-time τ_T . However, we realized by integrating with respect to ϵ_{\perp} instead of ϵ_{\parallel} that we are actually exactly probing the traversal time of the one dimensional motion through the barrier with a good control of approximations without any fit of the transmission (see Methods in the new manuscript). We thank the reviewer for triggering this improved data analysis and we believe that this new calculation provides real added value.

5. There is a number of places where language could be improved. See for example “peculiar” after Eq. (14) which should rather be “specific”? There are also some small grammar mistakes in the text.

This has been corrected throughout the manuscript to the extent possible.

6. In Fig. 1 the authors call their setup a turnstile. This is a bit misleading, since a turnstile leads to a clocked transmission due to a time-dependent driving. This has been corrected in lines 1-2 of the caption of Fig. 1.

3 Answers to Reviewer #3

We would like to thank the reviewer for his/her careful reading of the manuscript. We are glad he/she appreciated our work. He/she provided a main suggestion to improve the manuscript: “... for the readership of Nature Communications the given explanations appear quite complex and in the present form it may be better suited for a more specialized audience. Thus, I may only recommend publishing the manuscript in Nature Communications if the authors can enhance the accessibility of the manuscript also for non-specialists in the field.”

We agree with the reviewer that, our article should address to both electronic quantum transport and plasmonics communities whereas, at first sight, it seems to be mainly aimed at the quantum transport community investigating the Landauer-Büttiker formalism and their traversal-time. To address this issue, we have changed the introduction by citing the most convincing experiences that measure the traversal-time. These experiments deal with the tunneling time in atom ionization. For this purpose, we have changed the introduction (lines 21-24) and have added references [3,4,5] in the new manuscript. We have also added several references [12,13,14,31,32,45,47] concerning the inelastic photon emission by conductor whether it is junctions prepared by electromigration or formed between a biased STM-tip and metallic sample (see lines 40-45). This allows us to insist on the usefulness of our contribution in terms of transport properties of mesoscopic conductors. As reviewer kindly points out, we would like to stress that our study allows us not only to estimate the quantum efficiency of metallic tunnel junctions as a light emitter (see lines 200-209), but also to discuss the issue of the elastic vs. inelastic photon emission recently discussed in reference [45] (lines 211-214).

4 Answers to Reviewer #4

We would like to thank the reviewer for his/her careful reading of the manuscript. We are glad he/she pointed out that the traversal-time “seems to be new to me and may be an important step forward.”. He/she provided detailed comments and suggestions to greatly improve the manuscript. Below are our answers to his/her suggestions.

1. It is pointed out how the Fluctuation-Dissipation relation does not fit the results well. This does not come as a surprise since this is a highly non-equilibrium situation.

The reviewer has pointed out that there is no reason to compare our experimental results at optical frequencies and the fluctuation-dissipation relation (FDR) given by Eq. (1). We indeed have shown that the FDR breaks down because of the energy and voltage dependence of the transmission. However, we think it is important to compare our experimental results

FIG. 7. The solid line is an experimental conductance-voltage plot for an Al-I-Al junction. The barrier parameters $d=16.1$ Å. $\phi_1=2.03$ V, $\phi_2=0.73$ V were determined as outlined in the text. The points are calculated using these parameters and the diffuse barrier model (1). The dashed line is the mean of the two voltages required to produce a given experimental conductance.

Figure 1: “The solid line is an experimental conductance-voltage plot for an Al-I-Al junction. The barrier parameters $d \sim 1.61$ nm $U \sim 1.38$ eV and $\Delta U \sim 0.65$ eV were determined as outlined in the text. The points are calculated using these parameters and the diffuse barrier model (1). The dashed line is the mean of the two voltage required to produce a given experimental conductance.” The image has been taken from reference [34] in the new manuscript.

and the Landauer-Büttiker (LB) approach with the FDR. This relation is indeed largely used in the community working on quantum noise and the wide-ranging applications of the FDR are often highlighted. For example, reference [17] of the new manuscript states in page 5: “We call the relation (3.10) nonequilibrium fluctuation-dissipation theorem because of its general validity (we recall that no assumptions on geometry or interactions were made).... The current is not necessary linear in $\Delta\mu$ (the case of tunneling into a Luttinger liquid is an obvious example)”. Ref. [18] of the new manuscript states in page 1: “In this paper, we show that the FDR between the finite frequency quantum noise and dc nonequilibrium current is valid independently of the details of the conductor as well as of its environment provided the following hypotheses are satisfied: (i) validity of perturbation theory with respect to tunneling operators, (ii) vanishing of the dc current at zero voltage, and (iii) initial thermal equilibrium of the circuit in the limit of vanishing tunneling.” That is why we think it is important to demonstrate that the voltage-independent transmission is a crucial hypothesis for the FDR remains valid (see Methods).

2. The barrier is fabricated as symmetric L-R. Although this is the case, the model have to include a very large asymmetry parameter ΔU on the order of the barrier itself, U to fit the experiments. On the other hand the charges on the L, R electrodes are assumed to be the same. This would seem reasonable if there were no signatures of asymmetry but this is not the case.

Although the electrodes are identical (same metal and same thickness), such non-symmetrical characteristics has already been observed in the literature (see Fig. 1 of this letter, Brinkman *et al.* (1970)). The reported value of ΔU is not as high as our measured value. However we should note that the thickness of our electrodes is very small (5 nm). We have indeed carried out an exhaustive study of the effect of the electrode thickness (see Fig. 2 of this letter). Our measurements are in agreement with those of Brinkman *et al.* and we have observed that, the thinner the electrodes, the stronger the dissymmetry (see Fig. 1 of this letter). This effect can be explained by the high roughness of the thin electrodes and the fact that the lower one grows on the silicon oxide substrate while the upper one grows on the aluminum oxide of the first electrode which necessarily induces a dissymmetry.

Figure 2: Barrier height U and asymmetry ΔU for different electrodes thickness.

3. The model assumes completely planar surfaces such that the roughness is on a larger scale than the Fermi wavelength and transverse channels can be defined by splitting the free-electron energy expression. Is this justified? It is stated that there are no pin-holes but there could be roughness which might explain the high asymmetry cf. point 1.

We agree with the reviewer that the uniform transmission hypothesis is a strong assumption. However insulating barrier parameters in tunnel junctions such as barrier thickness and barrier height are often derived from a fitting procedure of the experimental $I(V)$ data with theoretical WKB model using a uniform trapezoidal barrier (see reference [35] in the new manuscript). Moreover, even if the effective thickness found using this approach is different from the structural one (see Ref. [34] with transmission electron microscopy, in the new manuscript), the obtained reasonable values prove that tunneling dominates the electronic conduction. Furthermore, if the distribution of transmission is assumed to be sufficiently homogeneous, conductance is expected to be dominated by the distribution of large transmissions. Combining the optical and the electronic transport measurement, we demonstrate that transmission is homogeneous on the nanoscopic, the microscopic and the macroscopic length scale:

- Nanoscopic length scale: the radio frequency shot noise is linear with the current (see Fig. 4 and lines 148-151 in the article) excluding any presence of pinholes. We come to the same conclusion by looking at the superconducting-insulator-superconducting (SIS) characteristic of the same kind of $Al/AlO_x/Al$ junction at low temperature when aluminum electrodes become superconducting. Fig. 3a in this letter indeed exhibits an infinite resistance for $|eV| < 2\Delta$ at 10 mK excluding any 1D well transmitted channel and so any pinhole.
- Microscopic length scale: fig. 4(a) shows the light intensity directly emitted by the plasmon diffusion on the roughness surface of junction. This indicates that electron-to-photon conversion in the tunnel junction is also homogeneous according the optical resolution of $\sim 5\mu\text{m}$. This excludes dramatic variations in barrier parameters as we can see in Fig. 4(a) which shows the cross-sections of the normalized light power along the two electrodes. We indeed observe an enhancement of the emission by the diffusion of surface plasmons on the sharp lateral step of 5 nm between the upper and the lower electrode (see Fig. 4(b)).
- Macroscopic length scale: at low temperature, the $Al/AlO_x/Al$ tunnel junction displays a supercurrent characterized by a critical current I_c (see Fig. 3a). When a

Figure 3: (a) $I(V)$ characteristic of a $\sim 1\text{ mm} \times \sim 0.5\text{ mm}$ $\text{Al}/\text{AlO}_x/\text{Al}$ tunnel junction measured at $T = 0.4\text{ K}$. The measured critical current is given by $I_c \sim 22\mu\text{A}$ and the Aluminum gap $\Delta \sim 165\mu\text{V}$. (b). Fraunhofer diffraction pattern of the critical current as a function of the magnetic flux through the barrier.

magnetic field H is applied parallel to the junction, the local value of the Josephson current oscillates with position: $j(x) = j_c \sin(\text{cte} + 2\pi H \times (2\lambda + d) \times L \times x/\phi_0)$ where λ is the penetration length, d the thickness of the barrier, L its length and $\phi_0 = h/e$ the flux quantum. If the local transmission is uniform then the local super current j_c is also uniform and the critical current is expected to show a Fraunhofer diffraction pattern $I_c/I_0 = |\text{sinc}(\pi\phi(H)/\phi_0)|$ where $\phi(H) = H \times (2\lambda + d) \times L$ is the magnetic flux through the barrier. Fig. 3b in this letter shows such a pattern proving the homogeneity on the macroscopic length scale $L \sim 1\text{ mm}$. The slight difference between theory and data points for $|\phi(H)| > \phi_0$ can be attributed to the homogeneity of λ in the electrodes.

Given that the inelastic photon emission is also a direct consequence of the tunneling (see Eqs. (3a-b)), the use of the simple homogeneous trapezoidal barrier model to describe the experimental $I(V)$ should be sufficient to describe the photon emission. According to reference [34], the thickness is roughly homogeneous ($\sim 2\text{ nm}$) and the flatness of the barrier fluctuates of $\sim 2\text{ nm}$ over a characteristic length of $\sim 50\text{ nm}$. The roughness of the barrier is therefore very small and justifies our planar surface hypothesis (see Eq. (7) for the radiation impedance). The estimated value of this impedance is actually in good agreement with the measured one (lines 171-176).

Figure 4: **(a)** Emitted light from the tunnel junction ($I = 1.7$ mA) directly observed with a sensitive camera in the spectral range $0.4 - 1 \mu\text{m}$. Red and blue dash lines correspond to cross-sections of the optical power along the x and y directions. **(b)** Normalized light power emitted in the spectral range $0.4 - 1 \mu\text{m}$ along the x and y directions. The upper (lower) electrode corresponds to the electrode in the y (x) direction.

4. The values of U , and ΔU should be justified. Are these reasonable/understandable based on the materials?

The value of $\Delta U \sim 2.9$ eV has been explained by the roughness of the thin electrode (see Fig. 2). Concerning the barrier height $U \sim 2.7$ eV, it is in reasonable agreement with the *ab initio* simulations of reference [7] of the new Supplemental Material):

$$U \sim E_g/4 \sim 2 \text{ eV},$$

where E_g is Alumina band-gap. One has to keep in mind that U can be slightly over-estimated because of the inhomogeneity of the junction parameters.

5. It is unclear to me that the traversal time generalizes from 1D to 3D in this way. Do you have a reference for the multi-channel case?

We agree with the reviewer that the quantity measured by Eq. (8) was not clearly defined. Our first idea was that 1D and 3D transmission in the tunneling limit are almost identical. This is for example shown in reference Phys. Rev. Lett. 95, 257001 (2005) where the authors measure the d-Wave Gap Symmetry in $YBa_2Cu_3O_7$ superconducting films by measuring the angle-resolved electron tunneling through oriented 3D tunnel junctions.

However, we have overcome this problem in the revised manuscript by clarifying the 3D calculation. We have added a remarque in lines 182-183 in the new manuscript to emphasize that our calculation only estimates the one dimensional traversal-time given by the WKB approximation. In the previous version of the manuscript, we have used the single mode transmission $\mathcal{T} = M/\epsilon_F \int_0^\epsilon T_{WKB}(\epsilon - \epsilon_\parallel, eV) d\epsilon_\parallel$ to give an estimate of the two dimensional traversal-time τ_T without a clear justification. However, we realized by integrating with respect to ϵ_\perp instead of ϵ_\parallel that we are actually probing the traversal-time of the one dimensional motion through the barrier with a good control of approximations without using any fit of the transmission. We indeed end-up with spectral expressions involving

the one dimensional WKB transmission $T_{WKB}(\epsilon, eV)$ and a quasi-distribution functions $\tilde{f}(\epsilon) = -\frac{Mk_B T}{\epsilon_F} \ln(f(-\epsilon))$ which corresponds to the integration of $f(\epsilon_{\perp} + \epsilon_{\parallel})$ over ϵ_{\parallel} (see Methods in the new manuscript):

$$S_{ii}(eV, h\nu) = \frac{2e^2}{h} \left\{ (1 + N(eV - h\nu)) \int d\epsilon_{\parallel} \left(\frac{\sqrt{T_{WKB}(\epsilon_{\parallel}, eV)} + \sqrt{T_{WKB}(\epsilon_{\parallel} + h\nu, eV)}}{2} \right)^2 \right. \\ \left. \left[\tilde{f}(\epsilon_{\parallel}) - \tilde{f} \left(\epsilon_{\parallel} + e \left(V - \frac{h\nu}{e} \right) \right) \right] \right. \\ \left. + N(eV + h\nu) \int d\epsilon_{\parallel} \left(\frac{\sqrt{T_{WKB}(\epsilon_{\parallel}, eV)} + \sqrt{T_{WKB}(\epsilon_{\parallel} - h\nu, eV)}}{2} \right)^2 \right. \\ \left. \left[\tilde{f}(\epsilon_{\parallel}) - \tilde{f} \left(\epsilon_{\parallel} + e \left(V + \frac{h\nu}{e} \right) \right) \right] \right\}.$$

6. The value of the traversal time estimate (1.1 fs) is on the very low part of the expected values for the model (Fig. 5). Is it clear why this is not inside the expected area (gray)?

We made clarifications about the quantity we measure thanks to the integration with respect to ϵ_{\perp} . By evaluating the generalized Fano factor \tilde{F} at voltage $eV = 2h\nu$, we directly estimate the traversal-time corresponding to the one dimensional motion through the barrier:

$$\tau_T(\epsilon, eV) = \int_{z_{min}(eV, \epsilon)}^{z_{max}(eV, \epsilon)} \sqrt{\frac{m}{2(U(z) + eV(1 - \frac{z}{d}) - \epsilon)}} dz. \quad (2)$$

However, we still have to determine the energy and the voltage at which we are measuring it. According to Eq. (17), the calculation of \tilde{F} allows us to get ride of the unknown transmission whose integration spans from ϵ_F to $\epsilon_F + h\nu$. Thus, the generalized Fano factor $\tilde{F}(eV, h\nu) = S_{ii}(eV, h\nu)/(e|I(V)|)$ evaluated at $V = \pm 2h\nu/e$ can be expressed in the zero temperature limit as a function of $\tau_T(\epsilon \sim h\nu/2, eV = 2h\nu)$ (see Fig. 6 and lines 162-163). We then conclude that we are measuring $\tau_T(eV = 2h\nu, \epsilon = h\nu) \sim 1.2$ fs. Our experimental result $\tau_T \sim 1.1$ fs is in very good agreement with this theoretical value.

We hope that these clarifications fully satisfy the reviewers. If we have misunderstood his/her point, we will be of course happy to further discuss it upon request. I look forward to your feedback on the revised manuscript.

Sincerely,

Julien Gabelli

Laboratoire de Physique des Solides,
CNRS, Univ. Paris-Sud, Université Paris-Saclay,
91405 Orsay Cedex, France
julien.gabelli@u-psud.fr

Reviewers' comments:

Reviewer #2 (Remarks to the Author):

I am satisfied with most of the answers of the authors and also see that they answered to the other referees' concerns in detail and rigorously. I therefore continue having a very positive opinion about this work. There are three different issues that I would nevertheless like to rise:

The first one is minor and was already part of the first round of communication. I am a bit confused about the use of the term "direct measurement" and would like to ask the authors to either very clearly define what they mean by this or to replace this expression by "fitting-free" or "from a model based on measurable quantities, only". I found this most confusing in the abstract (second-last line), line 29 and 55 in the introduction, and line 235 in the discussion.

The second issue is probably a matter of taste: the text is still pretty technical and it might be difficult to access for a non-expert. Since however a lot of background material is given in the supplementary and in references, I do not find this too much a hurdle. Nonetheless I want to point this out, since I can definitely notice a difference with respect to other Nature Comm. publications.

Finally, the third comment is more technical. I think that the discussion of the channel-averaging is not performed in a very transparent way. First of all the authors state in at least two places that they can choose whether to average over the transversal or longitudinal component. This is confusing, since this is just a matter of a mathematical transformation and not physically equivalent! However, also the technical derivation in the supplementary is not easy to follow:

- In Eq. (3) summation and integration appear in the wrong order. This is important since ϵ depends on k_{\perp} .
- In the summation index k_{\perp} needs to be a vector
- It is not clear how the denominator in the central expression of Eq. (4) could be replaced by a Fermi-energy. This should be ϵ_{\perp} .
- It is fully unclear, where in this same set of equations the condition $\epsilon = \epsilon_{\perp} + \epsilon_{\parallel}$ enters. Or, if it is not entering, why it disappeared.
- Also, why are the factors in the integrand of Eq. (3) averaged independently of each other? I guess these aspects have to be clarified and/or adjusted.

Reviewer #3 (Remarks to the Author):

The authors have addressed my earlier comment and implemented further clarifications and explanations to the manuscript. In particular, they significantly increased the readability of the manuscript also for a non-specialized audience. Thus, they also clarify the impact of their findings with respect to transport properties of mesoscopic conductors aiming at a broader audience. Thus, I recommend publication of the article in Nature Communications in its present form.

Reviewer #4 (Remarks to the Author):

I am satisfied with all answers except I am missing at least comments on the STM noise work, cf. my question:

Moreover, is this work providing a substantial contribution beyond the more STM-type experiments (e.g. PRL 109, 186691 (2012), not cited) where the channels are defined by a few molecular orbitals?

We thank the reviewers for their helpful comments, that have helped us to improve our manuscript. Below, we provide a detailed response to each of them. The changes made to the original manuscript appear in red in the manuscript.

1 Answers to Reviewer #2

I am satisfied with most of the answers of the authors and also see that they answered to the other referees concerns in detail and rigorously. I therefore continue having a very positive opinion about this work.

We thank the reviewer for acknowledging the improvements made in this manuscript and for his/her positive opinion.

There are three different issues that I would nevertheless like to rise:

1. The first one is minor and was already part of the first round of communication. I am a bit confused about the use of the term “direct measurement” and would like to ask the authors to either very clearly define what they mean by this or to replace this expression by “fitting-free” or “from a model based on measurable quantities, only”. I found this most confusing in the abstract (second-last line), line 29 and 55 in the introduction, and line 235 in the discussion.

The Reviewer is right. The term “direct measurement” is in fact not appropriate and we apologize for this miswording. We have thus modified our wording when necessary. In particular, we have changed the wording “direct measurement” to “estimation” whenever it was used. We have also used the formulation suggested by the referee in the discussion “*give an approximative value of the traversal-time by fitting-free model*”.

2. The second issue is probably a matter of taste: the text is still pretty technical and it might be difficult to access for a non-expert. Since however a lot of background material is given in the supplementary and in references, I do not find this too much a hurdle. Nonetheless I want to point this out, since I can definitely notice a difference with respect to other Nature Comm. publications.

The referee raises the issue of the technical level of our text. We would like to argue that our paper is actually at the intersection of different domains and therefore mixes multiple concepts and experimental techniques, which necessary implies technical parts. Nevertheless, we would like to note the substantial efforts we have already made for the first round of revisions to increase readability, and both Reviewer#2 and #3 acknowledge the gain in clarifications.

3. The third comment is more technical. I think that the discussion of the channel-averaging is not performed in a very transparent way. First of all the authors state in at least two places that they can choose whether to average over the transversal or longitudinal component. This is confusing, since this is just a matter of a mathematical transformation and not physically equivalent! However, also the technical derivation in the supplementary is not easy to follow:

- In Eq. (3) summation and integration appear in the wrong order. This is important since ϵ depends on k_{\perp} .
- In the summation index k_{\perp} needs to be a vector
- It is not clear how the denominator in the central expression of Eq. (4) could be replaced by a Fermi-energy. This should be ϵ_{\perp} .
- It is fully unclear, where in this same set of equations the condition $\epsilon = \epsilon_{\perp} + \epsilon_{\parallel}$ enters. Or, if it is not entering, why it disappeared.
- Also, why are the factors in the integrand of Eq. (3) averaged independently of each other?

We would like to thank the referee for his/her careful reading of the manuscript and pointing out these technical issues.

The reviewer is correct that the summation and the integration in Eq. (3) of the supplementary appear in the wrong order. We apologize for this mistake and we have corrected it.

We also recognize that the hypothesis of the model were unfortunately not sufficiently defined. We follow the Simmons' model (ref [4] of the supplementary) where electrons are assumed to be free particles which have no interaction with each other and with the lattice. It allows us to describe electrons by one-electron wave function with a wave vector \mathbf{k} and an energy ϵ :

$$\epsilon = \frac{\hbar^2}{2m} |\mathbf{k}|^2 = \frac{\hbar^2}{2m} (\mathbf{k}_\perp^2 + \mathbf{k}_\parallel^2) \quad (1)$$

A free electron moves in an energy potential which only depends on longitudinal direction z . The Schrödinger equation can be splitted into three independent 1D problems: one for each dimension. The total wave function is then the product of one-dimensional solutions. The solutions for transversal direction (x and y direction) have the form $\psi_\perp^\pm = \exp(\pm i \mathbf{k}_\perp \cdot \mathbf{r}_\perp)$, which are independent of z . This is consistent with the conservation of the component of the vector parallel to the barrier. The solution in the z direction depends on the effective potential U_{eff} . The integral is a summation over all electrons in k -space times the electron group velocity, $(1/\hbar)\partial\epsilon/\partial k_z$, weighted by the tunnel probability and the appropriate Fermi-Dirac functions at that energy. Assuming that the transverse dimensions are much larger than the Fermi wavelength, the quantization in the transverse direction can be neglected and k_\perp is continuous. The integration over k_\perp is thus restricted to values which conserve ϵ and k_\perp . By using the free-electron density of states and the Wentzel-Kramers-Brillouin (WKB) transmission coefficient the total tunneling current is obtained:

$$I(V) = \frac{Se}{2\pi^2\hbar} \int_0^\infty d\epsilon [f(\epsilon) - f(\epsilon + eV)] \int d\mathbf{k}_\perp \mathcal{T}_{3D}(\epsilon, V). \quad (2)$$

For free electron dispersions the transmission probability only depends on the kinetic energy along the z direction ϵ_\parallel . Averaging over all possible values of ϵ_\parallel for an incoming electron with energy ϵ , we obtained its mean transmission :

$$\overline{\mathcal{T}}_{3D}(\epsilon, V) = \frac{1}{\epsilon} \int_0^\epsilon T_{WKB}(\epsilon_\parallel, eV) d\epsilon_\parallel. \quad (3)$$

We have accordingly modified the lines 52-75 of the supplementary material.

2 Answers to Reviewer #3

The authors have addressed my earlier comment and implemented further clarifications and explanations to the manuscript. In particular, they significantly increased the readability of the manuscript also for a non-specialized audience. Thus, they also clarify the impact of their findings with respect to transport properties of mesoscopic conductors aiming at a broader audience. Thus, I recommend publication of the article in Nature Communications in its present form.

We are grateful to this reviewer for acknowledging the improvements made in our manuscript and for recommending its publication in Nature Communications.

3 Answers to Reviewer #4

I am satisfied with all answers except I am missing at least comments on the STM noise work, cf. my question: Moreover, is this work providing a substantial contribution beyond the more STM-type experiments (e.g. PRL 109, 186691 (2012), not cited) where the channels are defined by a few molecular orbitals?

We thank the reviewer for recognizing the improvements made in our revision.

We have added the reference suggested by the reviewer (ref [14]). We believe that our work substantially improve the understanding of light emission by tunnel junctions in a way that is not possible with STM experiments. The main argument is that the heating of the electrons observed in these systems make their quantitative understanding difficult. In our experiment, we have carefully checked that no heating occurs. Furthermore, our analysis of the traversal time is new, and should be of interest to a broad community.

We hope that these clarifications will fully satisfy all reviewers. We look forward to their feedback on this revised manuscript.

Sincerely,

Julien Gabelli
aboratoire de Physique des Solides, CNRS, Univ. Paris-Sud, Université Paris-Saclay,
91405 Orsay Cedex, France
julien.gabelli@u-psud.fr

REVIEWERS' COMMENTS:

Reviewer #2 (Remarks to the Author):

I would now recommend the paper for publication.

I would however recommend to the authors to reconsider the supplementary section about the energy-dependent transmission once again, even though it is of course "only" a supplementary material. These are the typos/problems, I spotted:

- It is confusing that "z" and "parallel" are used at the same time and interchangedly. Wouldn't it be possible to replace all \parallel -supscripts by z? Both for ϵ_{\parallel} and k_{\parallel} ? This would definitely be more readable.
- In line 65, you write "The integral is a summation...". I assume that what you mean is "The integral in Eq. (3) is a summation...".
- The different transmission probabilities are sometimes functions of "V" and sometimes functions of "eV". It would be nice to treat this consistently.
- Much more important is the fact that T_{3D} , \bar{T}_{3D} , T , and \bar{T}_{3D}/M can not be distinguished from each other and are probably partly the same thing... In Eq. (5): Is this \bar{T}_{3D} ? The number M that you defined actually does not occur in any of the equations, etc.
- minor: In line 59, "end" should be "and"

Reviewer #4 (Remarks to the Author):

I am satisfied with the answers to my own points as well as the points raised by the other referees. I recommend publication.

We would like to thank the reviewers #2 and #4 for their last reading. Below, we provide a detailed response to reviewer #2.

1 Answers to Reviewer #2

I would however recommend to the authors to reconsider the supplementary section about the energy-dependent transmission once again, even though it is of course “only” a supplementary material.

There are three different issues that I would nevertheless like to rise:

1. It is confusing that “z” and “parallel” are used at the same time and interchangeably. Wouldn't it be possible to replace all parallel-supscripts by z? Both for ϵ_{\parallel} and k_{\parallel} ? This would definitely be more readable.

We agree with the reviewer. We thus have checked the calculations by clearly writing the link between the 3D transmission $T_{3D}(k_x, k_y, \epsilon)$ and the WKB transmission $T_{\text{WKB}}(\epsilon_{\parallel}, eV)$ (supplementary equation (5)). We also clearly compute the change of variables and the continuous boundary ($\int d\epsilon \sum_{k_x, k_y} \rightarrow \int d\epsilon_{\parallel} \int_{k_{\perp}}$) by showing the transverse modes M of the 3D junction (supplementary equation (6)). We thus have decided to keep the parallel-supscripts to define the magnitude of the transverse wave vector $\mathbf{k}_{\parallel} = (k_x, k_y)$. This also allows us to have a notation consistent with the equations of the article.

2. In line 65, you write “The integral is a summation...”. I assume that what you mean is “The integral in Eq. (3) is a summation...”.

The referee is correct : we have clarified the transition to the continuous limit (supplementary equation (6)) on page 5.

3. Much more important is the fact that T_{3D} , \bar{T}_{3D} , T , and \bar{T}_{3D}/M can not be distinguished from each other and are probably partly the same thing... In Eq. (5): Is this \bar{T}_{3D} ? The number M that you defined actually does not occur in any of the equations, etc.

Thanks to the change of variable (integrate over the transversal energy), we introduce the mean value of the transmission of the junction channel (supplementary equation (9)) and estimate the number of conducting channels $M \sim 2.4 \times 10^{11}$. The average transmission of a tunneling channel is then smaller than 1.7×10^{-11} proving the tunneling limit.

4. In line 59, “end” should be “and”

It has been corrected.

We would like to thank again the referee for his/her careful reading of the manuscript and pointing out these technical issues.

Sincerely,

Julien Gabelli
Laboratoire de Physique des Solides rue Nicolas Appert
Bâtiment 510
91405 Orsay Cedex - France
julien.gabelli@u-psud.fr